# Urinary leukotrienes and histamine in patients with varying severity of acute dengue

Tehani Silva[1,2], Chandima Jeewandara[1], Laksiri Gomes[1], Chathurika Gangani[1], Sameera D. Mahapatuna[1], Thilagaraj Pathmanathan[1], Ananda Wijewickrama[3], Graham S. Ogg[4], Gathsaurie Neelika Malavige [1,4]*

1 Centre for Dengue Research, University of Sri Jayewardenepura, Nugegoda, Sri Lanka, 2 General Sir John Kotelawala Defence University, Rathmalana, Sri Lanka, 3 National Institute of Infectious Diseases, Angoda, Sri Lanka, 4 MRC Human Immunology Unit, MRC Weatherall Institute of Molecular Medicine, University of Oxford, Oxford, United Kingdom

* gathsaurie.malavige@ndm.ox.ac.uk

**Data Availability Statement:** All relevant data are within the manuscript and figure files.

**Funding:** The funding was provided by the Centre for Dengue Research, University of Sri

## Abstract

### Background

Vascular leak is a hallmark of severe dengue, and high leukotriene levels have been observed in dengue mouse models, suggesting a role in disease pathogenesis. We sought to explore their role in acute dengue, by assessing levels of urinary LTE4 and urinary histamine in patients with varying severity of acute dengue.

### Methods

Urinary LTE4, histamine and creatinine were measured by a quantitative ELISA, in healthy individuals (n = 19), patients with dengue fever (DF = 72) and dengue haemorrhagic fever DHF (n = 48). The kinetics of LTE4 and histamine and diurnal variations were assessed in a subset of patients.

### Results

Urinary LTE4 levels were significantly higher (p = 0.004) in patients who proceed to develop DHF when compared to patients with DF during early illness (≤ 4 days) and during the critical phase (p = 0.02), which continued to rise in patients who developed DHF during the course of illness. However, LTE4 is unlikely to be a good biomarker as ROCs gave an AUC value of 0.67 (95% CI 0.57 and 0.76), which was nevertheless significant (p = 0.002). Urinary LTE4 levels did not associate with the degree of viraemia, infecting virus serotype and was not different in those with primary vs secondary dengue. Urinary histamine levels were significantly high in patients with acute dengue although no difference was observed between patients with DF and DHF and again did not associate with the viraemia. Interestingly, LTE4, histamine and the viral loads showed a marked diurnal variation in both patients with DF and DHF.

Jayewardenepura (GNM), National Science
Foundation, Sri Lanka (RPHS/2016/D-06, GNM),
The funders had no role in study design, data
collection and analysis, decision to publish, or
preparation of the manuscript.

**Competing interests:** The authors have declared
that no competing interests exist.

## Conclusions

Our data suggest that LTE4 could play a role in disease pathogenesis and since there are
safe and effective cysteinyl leukotriene receptor blockers, it would be important to assess
their efficacy in reducing dengue disease severity.

## Introduction

Dengue infections are one of the most rapidly emerging mosquito-borne viral infections in the
world. As 70% of dengue infections occur in Asia, dengue causes a huge burden to resource-
poor economies in developing countries [1], with10.5 million individuals being admitted hos-
pital annually [2]. Approximately 26% to 33.6% of hospitalized patients have shown to develop
severe disease manifestations such dengue haemorrhagic fever (DHF) and organ impairment,
while 0.6% to 1.5% require admission to the intensive care units [3–5]. Although the recent
case fatality rates in many countries have declined to <0.5% due to intense monitoring and
meticulous fluid management, the case fatality rates are still around 2.6% in some countries
such as India [6]. Therefore, there is an urgent need to develop biomarkers which can identify
patients who will develop severe disease and effective drugs to treat dengue.

Endothelial dysfunction leading to vascular leak in the hallmark of DHF [7]. Plasma leakage
leads to accumulation of fluid in pleural and peritoneal cavities, and when severe, it causes
reduction of blood pressure leading in poor organ perfusion resulting in shock and organ dys-
function. Many inflammatory cytokines, lipid mediators and secretory virus protein NS1 has
been shown to contribute to the vascular leakage [8–11]. The dengue virus (DENV) has been
shown to activate mast cells, and the vascular leak in dengue mouse models has been shown to
be mast cell dependent [12]. Mast cell stabilizing drugs such as cromolyn and ketotifen and the
leukotriene inhibitor montelukast, significantly reduced the vascular leak in dengue mouse
models, implying their role in dengue pathogenesis [12].

We have previously shown that many inflammatory lipid mediators such as platelet activat-
ing factor (PAF) and secretory phospholipase A2 levels were higher in patients with DHF, and
that endothelial dysfunction induced by dengue sera was inhibited by PAF-R blockers, in
vitro, which also showed a tend towards a reduction in vascular leakage in patients with acute
dengue [3, 9, 13]. Leukotrienes are a group of inflammatory lipid mediators that are produced
by the action of lipoxygenase enzymes on arachidonic acid substrates, that lead to production
of LTB4 which metabolize into LTC4, LTD4 and LTE4 [14]. These mediators are potent che-
moattractants to neutrophils and eosinophils, and some are also known to increase vascular
permeability and induce vascular leak [14, 15]. LTE4, has been shown to induce vascular leak,
by contracting the endothelial cells in the post-capillary venules in a dose-dependent manner
[16]. Serum LTB4 levels, which were assessed in a small cohort of patients with acute dengue
infection, showed that it was elevated in patients with acute dengue when compared to healthy
individuals and correlated with inflammatory markers such as highly sensitive CRP [17]. How-
ever, if leukotrienes correlate with clinical disease severity, viraemia and serostatus and the
relationship with the onset of vascular leak in acute dengue has not been investigated.

Currently all individuals who have warning signs of possible severe disease such as persis-
tent vomiting, abdominal pain, leucopenia along with thrombocytopenia are admitted to the
hospital [18]. However, only 25% to 34% of such patients would develop complications such as
DHF and organ dysfunction [3, 5, 19]. The onset of plasma leakage usually occurs during 3 to
6 days and lasts for 24 to 48 hours, which is known as the critical phase [7, 18]. Those who do

not develop vascular leak have a self-limiting illness known as dengue fever, and usually recover without any intervention, except if accompanied by bleeding manifestations. However, due to the absence of a reliable biomarker, all patients who are admitted should be closely monitored, several times a day, which is a huge burden in overcrowded hospital facilities with limited staff. Several potential biomarkers have been identified such as serum chymase, serotonin, VEGF and tryptase levels [20–23]. However, it would be useful to evaluate a urinary biomarker to predict the development of DHF during early illness as it would user friendly.

Urinary LTE4 is known to be a stable metabolite of LTC4 and LTD4 and therefore, an indicator of cysteinyl leukotriene activity [24, 25]. LTE4 is produced by many cell types such as mast cells, basophils, eosinophils, macrophages and neutrophils and is relatively stable in urine [26]. Urine and spot urine LTE4 levels have been evaluated as a diagnostic marker of atopic asthma in pre-school children [25, 27]. In this study we evaluated the levels of urinary LTE4 and histamine in a large cohort of patients with varying severity of acute infection during early illness, their kinetics throughout the illness and also diurnal variations. We investigated the relationship between the urinary LTE4 levels, with the onset of vascular leakage, viral loads and serostatus.

## Materials and methods

### Patients for analysis of LTE4 levels in early illness

120 adult patients with acute dengue infection were recruited from the National Institute of Infectious Diseases during the years 2018–2019, following informed written consent. The duration of illness at the time of recruitment and obtaining blood and urine samples was ≤ 4 days of illness in 120 patients and between 5 to 6 days in 54 patients. The first day of illness was considered as the first day they developed fever. Any individual who had chronic liver disease, chronic renal disease and pregnant women were excluded from the study. A subset of the patients (DF = 25 and DHF = 23) recruited with a duration of illness of ≤ 4 days were followed up daily to determine the kinetics of changes in urinary LTE4 (described below).

All patients were assessed several times a day for changes in blood pressure, urine output, presence of bleeding manifestations and possible leakage. None of the patients who were recruited with ≤ 4 days of illness had evidence of fluid leakage and they were only diagnosed to have DHF, during subsequent follow up, when fluid leakage was detected. Ultrasound scans were performed to determine the presence of fluid leakage in pleural and peritoneal cavities. All the clinical parameters (e.g. blood pressure, temperature, heart rate, pulse pressure and fluid leakage) and investigation results such as full blood count, liver function tests, serum electrolytes were recorded throughout the course of illness. The severity of dengue infection was classified according to the 2011 WHO dengue diagnostic criteria [18]. Accordingly, patients with a rise in haematocrit above ≥ 20% of the baseline haematocrit or with clinical or ultrasound scan evidence of plasma leakage were classified as having DHF. Based on this classification, in those recruited in ≤4 days of illness (n = 120), 72 patients had DF and 48 DHF and those recruited between day 5 to 6 (n = 54), 27 had DF and 27 had DHF. Nineteen healthy individuals who did not having any atopic illnesses (allergic rhinitis, asthma, food allergies or eczema), and who were not on antihistamines or steroids were recruited as healthy controls.

### Patients for analysis of serial changes in LTE4 levels and diurnal variations

In order to assess the kinetics of LTE4 throughout the course of illness, we obtained daily urine (9.00 am) and blood samples from 48/120 patients who were recruited on ≤ 4 days of illness (25 DF and 23 DHF), throughout the course of illness. In addition, as we earlier found that there was a diurnal variation in platelet activating factor (PAF) in 2 cohorts of patients [9,

[13], we also followed 22/120 patients who were recruited on ≤ 4 days of illness (14 who had DF and 8 with DHF), to assess possible diurnal changes in urinary $LTE_4$ levels in the morning (9 am) and the afternoon (3 pm). We could not obtain daily blood and urine samples (or twice daily samples) from the whole cohort of 120 patients who were recruited on ≤ 4 days of illness to explore the kinetics and diurnal variation of LTE4 and viral loads as the same cohort of patients did not consent to be bled twice a day due to the frequent bleeding of these patients (four times a day), as a part of their dengue management.

## Ethics approval

Ethical approval was obtained by the Ethics Review Committee of the Faculty of Medical Sciences, University of Sri Jayewardenepura (Ethics application number: 12.15). All patients were recruited following written informed consent.

## Quantification of urinary LTE4, histamine and creatinine

Levels of LTE4 and histamine were measured in urine stored at -80˚C, using leukotriene quantitative ELISA (Cayman Chemical, USA) and histamine quantitative ELISA (Abcam, UK) according to the manufacturer's instructions. Assays were performed in urine diluted at 1:5 ratio for assessment of leukotriene levels and undiluted urine for used for measurement of histamine. As the LTE4 and histamine levels can change according to the renal function, urinary creatinine levels were measured in all patients and healthy individuals in order to normalize the urinary LTE4 and histamine levels using the urinary creatinine colorimetric assay (Cayman Chemical, USA). These assays were carried out and interpreted according to the manufacturer's instructions.

## Determining viral loads and the DENV serotypes

Viral RNA from all serum samples were extracted using QIAamp Viral RNA Mini Kit (Qiagen, USA) according to manufacturer's instructions. Reverse transcription of the extracted viral RNA to cDNA was done using a high capacity reverse transcription kit (Applied biosystems, USA). To detect and quantify the DENV, a quantitative multiplex real time PCR was carried out as previously described (Fernando *et al.*, 2016), using the CDC real time oligonucleotide primers, and dual labeled probes for DENV 1–4 serotypes (Life technologies, USA). The reaction was performed for 20 seconds at 95˚C for initial denaturation, followed by 40 cycles, 95˚C at 3 seconds and 60˚C at 30 seconds. The threshold cycle value (Ct) for each reaction was determined by manually setting the threshold limit. A multiplex method was optimized to quantify the four serotypes in a single reaction and viral quantification of unknown samples was performed using the standard curve.

## Determining primary and secondary dengue

Dengue antibody assays were performed in 77 patients, who had a duration of illness of more than 5 days, using a commercial capture-IgM and IgG Enzyme- Linked Immunosorbent Assay (ELISA) (Panbio, Australia). Results were interpreted according to the manufacturer's instructions. According to the WHO 2011 criteria, patients with an IgM: IgG ratio of >1.2 were classified as having a primary dengue infection, while patients with IgM: IgG ratios <1.2 were categorized under secondary dengue infection. Based on these criteria 26 (34%) had a primary dengue infection and 51(66%) had a secondary dengue infection.

## Statistical analysis

Statistical analysis was performed using Graph pad PRISM version 8. The differences in LTE4 in single samples were done using the two tailed Mann-Whitney U-test and Kruskal-Wallis test. The degree of association between urinary LTE4 levels and viral loads, antibody levels and other laboratory tests was analyzed using Spearman correlation coefficient test. Changes in the LTE4 levels throughout the course of illness was compared using the Holm-Sidak method. Corrections for multiple comparisons were completed using the Holm-Sidak method and the significance value was set at 0.05 (alpha). The Wilcoxon paired signed rank test was carried out to determine significant of changes of LTE4, histamine and viral loads, in paired samples obtained in the morning and afternoon. A receiver-operator characteristic (ROC) curves showing the area under the curve (AUC) were generated to determine the discriminatory performance of LTE4 in predicting those who will develop DHF during early illness. The non-parametric Kruskal-Wallis (KW) test was used to analyze LTE4 levels in different age groups.

# Results

## Patient characteristics

A total of 120 patients with a duration of illness of day $\leq$ 4 were included in this study out of which 72 individuals had DF and 48 proceeded to develop DHF based on the 2011 WHO classification of dengue disease severity **(Table 1)**. Two patients who had DHF developed shock. All patients recovered and there were no fatalities. Six (8.3%) patients with DF and 7 (14.5%) patients with DHF had bleeding manifestations. In the patients with DF, four had vaginal bleeding and two had melena, whereas in the patients with DHF, one had melena and five had vaginal bleeding. Although 6 patients with DF had bleeding manifestations, none of them had evidence of fluid leakage and therefore, did not fulfill the criteria to be classified as DHF.

## Urinary LTE4 levels in patients with acute dengue

In patients with a duration of illness of $\leq$ 4 days (n = 120), the urinary LTE4 levels were significantly higher (p = 0.004) in patients who proceed to develop DHF (median 1741, IQR 980.1–2,799 pg/mg creatinine) when compared to patients with DF (median 1178, IQR 592.6–2,034 pg/mg creatinine) **(Fig 1)**. Although the LTE4 levels in patients with DF was higher than in healthy individuals (median 706.6 pg/mg creatinine, IQR 535.3–1639 pg/mg creatinine) this

**Table 1. Clinical and laboratory features of patients recruited for the study with ≤4 days of illness.**

| Clinical Characteristics | DF (n = 72) | DHF (n = 48) |
|---|---|---|
| Vomiting | 20 (27.8%) | 19 (39.6%) |
| Abdominal pain | 09 (12.5%) | 14 (29.2%) |
| Arthralgia | 54 (75%) | 44 (91.6%) |
| Myalgia | 45 (62.5%) | 39 (81.2%) |
| Diarrhoea | 19 (26.4%) | 22 (45.8%) |
| Bleeding manifestations | 06 (8.3%) | 07 (14.5%) |
| Ultrasound scan evidence of fluid Leakage (pleural effusions or ascites) | 00 (0.0%) | 48 (100%) |
| Lowest platelet count | | |
| <20,000cells/mm$^3$ | 01 (1.4%) | 23 (47.9%) |
| 20,000–50,000 | 11 (15.3%) | 21 (43.8%) |
| 50,000–100,000 | 32 (44.4%) | 04 (8.3%) |
| >100,000 | 28 (38.9%) | 00 (0.0%) |

difference was not significant (p = 0.148). The urinary LTE4 levels were also significantly high (p = 0.02) in patients with DHF (median 1755, IQR 1,102–2,875 pg/mg creatinine) on day 5 to 6 of illness compared to those with DF (median 1,263, IQR 729.9–2,383 pg/mg creatinine) (**Fig 1**).

## Changes in urinary LTE4 based on serostatus, viral loads and infecting virus serotype

It was previously shown that patients with secondary dengue infection had higher serum chymase levels and it was shown that this is likely to be due to increased mast cell degranulation by the DENV in the presence of DENV antibodies in mouse models [20, 28]. There was no significant difference (p = 0.25) in urinary LTE4 levels in patients with primary dengue (median 1242, IQR 654.9–1671 pg/mg creatinine) compared to those with secondary dengue (median = 1560, IQR 775.0- 2650pg/mg creatinine) (**Fig 2A**). However, patients with DHF during secondary dengue had significantly higher levels (p = 0.02), than in those with DF.

There was no association between the viral loads and urinary LTE4 levels in either patients with DF (Spearmans r = -0.09, p = 0.46), or DHF (Spearmans r = 0.18, p = 0.27) (**Fig 2B**). Of the 120 patients recruited in the study during early illness of which the infecting DENV serotype was known, 20 was infected with DENV 1, 61 with DENV 2 and 11 with DENV 3 (**Fig 2C**). There was no significant difference in the urinary LTE4 levels in patients infected with different DENV serotypes (**Fig 2C**). There was no association of urinary LTE4 levels with the white cell counts, although weak but significant inverse correlation was seen with the platelet counts (Spearman r = -0.19, p = 0.04)

## Usefulness of LTE4 in predicting development of DHF in early illness

As urinary LTE4 levels were higher in patients who proceeded to develop DHF during early illness ($\leq$ 4 days of illness), we sought to assess if urinary LTE4 could be used as a predictive

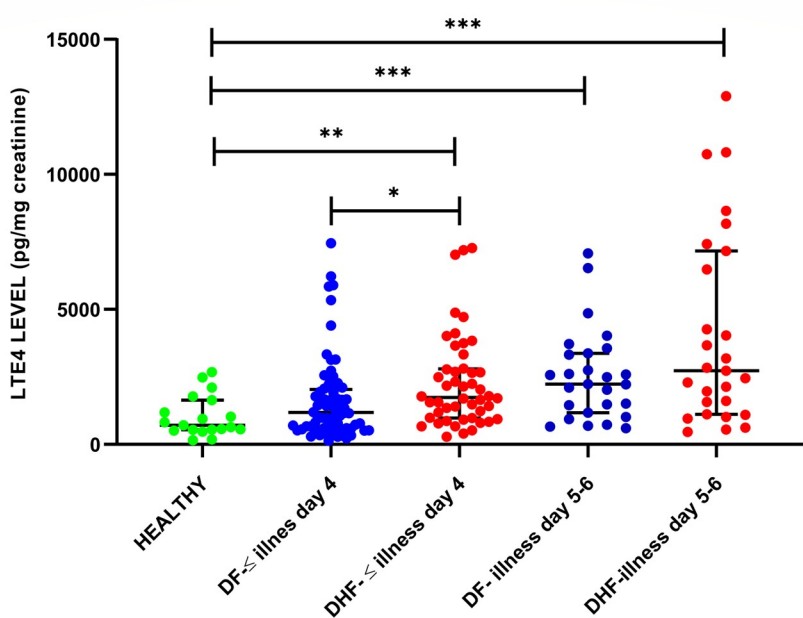

**Fig 1. Urinary LTE4 levels in patients with acute dengue.** Urinary LTE4 levels were measured by quantitative ELISA, in healthy individuals (n = 19), in those with DF (n = 72) and those who progressed to develop DHF (n = 48) with <4 days of illness and in patients with DF (n = 27) and DHF (n = 27) in 5 to 7[th] day of illness. The p values were calculated by the Mann-Whitney test. Error bars represent the median and IQR. *P <0.05, **P<0.001, *** P<0.0001.

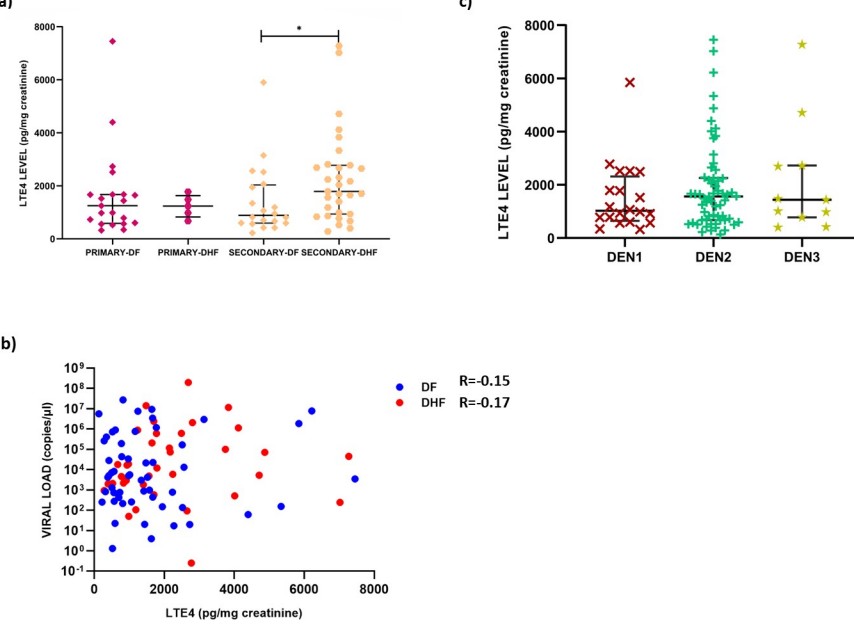

**Fig 2. Association of urinary LTE4 levels with serostatus, viral loads and infecting virus serotype in patients with acute dengue.** Urinary LTE4 levels were measured in patients with DF (n = 21) or DHF (n = 05) due to primary infection or DF (n = 20) or DHF (n = 22) due to secondary infection (a), urinary LTE4 levels were correlated with viral loads in patients with DF and DHF. (b), and the differences in the urinary LTE4 levels were assessed in patients infected with different virus serotypes DENV1 (n = 20), DENV2 (n = 61) and DENV3 (n = 11). The p values were calculated by the Mann-Whitney test and the correlation assessed by the Spearmans correlation. Error bars represent the median and IQR. *P <0.05.

marker of subsequent development of DHF in early illness. Receiver-operator characteristic (ROC) curves showing the area under the curve (AUC) were generated to determine the discriminatory performance of LTE4 for DHF in early illness, which showed an AUC value of 0.67, with a 95% confidence interval of 0.57 and 0.76, which was significant (p = 0.002) (**Fig 3**).

## Kinetics of changes of LTE4 throughout the course of illness

LTE4 by itself has shown to cause endothelial dysfunction leading to vascular leakage [16]. Since our results showed that urinary LTE4 levels were significantly higher in patients with DHF, during early illness, we sought to investigate if LTE4 levels were highest during the febrile phase or the critical phase to further understand the possible contribution of LTE4 to disease pathogenesis. In order to study the kinetics of LTE4, we measured the urinary LTE4 levels in 25 patients with DF and 23 patients who developed DHF throughout the course of illness with urine samples taken at 9am in the morning. We found that the urinary LTE4 levels rose in patients with DHF from day 3 to 5 and remained high whereas there was no such rise from day 3 in patients with DF (**Fig 4A**). The patients who developed vascular leak (those who developed DHF), entered the critical phase between day 4 and 5 of illness and the rise of LTE4 levels in these patients was when they entered the critical phase. Although the urinary LTE4 levels was high throughout the illness in patients with DHF compared to those with DF, this was not significant at any time point, probably due to the lower number of patients included in the serial sampling, which was likely not sufficiently powered to see changes of statistical significance.

We previously showed that levels of PAF showed a diurnal variation in two cohorts of patients with acute dengue [9, 13]. Patients with DHF, had high PAF levels in the morning,

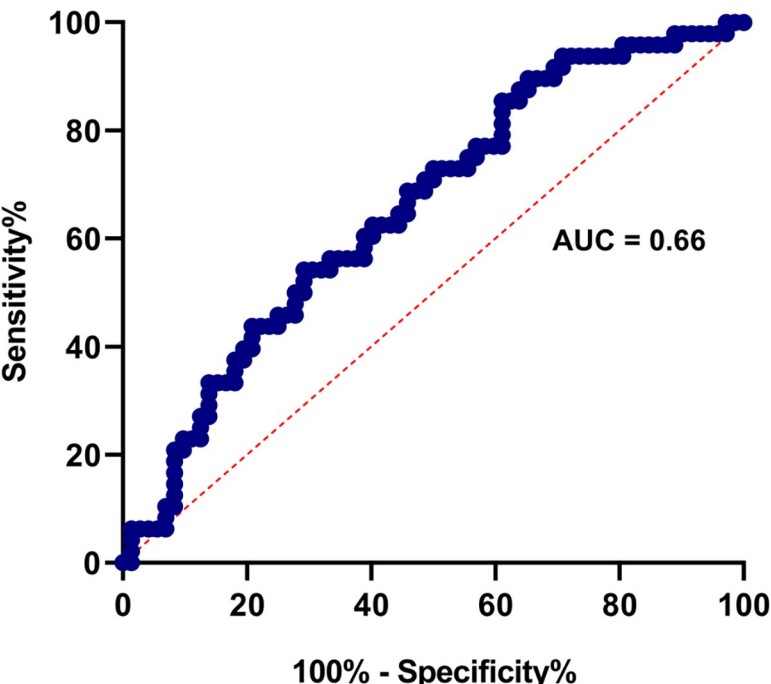

**Fig 3. The usefulness of urinary LTE4 as a predictive factor of DHF.** Receiver-operator characteristic (ROC) curves showing the area under the curve (AUC) were generated to determine the discriminatory performance of urinary LTE4 in predicting those who will develop DHF during early illness ($\leq$ 4 days) in those who progressed to develop DHF (n = 48) or DF (n = 72). Area under the curve AUC = 0.66 and P = 0.0039.

which markedly reduced in the afternoon[2]. Therefore, in order to find out if similar changes were observed with LTE4, levels were measured in the morning (9am) and late afternoon (3pm), in 14 patients with DF and 8 patients with DHF. We found that in some patients while LTE4 levels fell towards the afternoon, in others the LTE4 levels were higher at 3pm (**Fig 4B**). Such differences in urinary LTE4 levels were seen in both patients with DF and DHF (**Fig 4B**). Urinary LTE4 levels were higher in the afternoon samples in 71% (10/14) patients with DF and 50% (4/8) patients with DHF, when compared to the morning urinary LTE4 levels, but overall the timing differences were not significant.

In some patients, several fold changes in the viral loads were also seen between morning and afternoon samples in patients with both DF and DHF (**Fig 4C**). Viral loads were higher in the afternoon samples in 38% (5/13) patients with DF and 57% (4/7) patients with DHF, when compared to the morning viral loads. However, these changes in viral loads did not reflect in the changes in urinary LTE4 levels. For instance, while the LTE4 level increased the afternoon samples in 14 patients, the viral loads only increased in 9 samples (**Fig 4C**).

## Association of urinary LTE4 levels with age and gender in acute dengue

Of the 120 patients who were recruited during early illness ($\leq$ 4 days of illness), 48 (40%) were females and 72 (60%) were males. There was no significant difference in the LTE4 levels between males and females (p = 0.343) (**Fig 5A**). However there was a significant positive correlation between urinary LTE4 levels and age (Spearman r = 0.20 and p = 0.03) (**Fig 5B**). A significant difference (p = 0.003) was observed in the urine LTE4 levels in those who were $\leq$ 25 years (n = 55), those who were 26 to 40 years (n = 31) and those who were $\geq$ 41 years (n = 33) (**Fig 5C**). However, the proportion of those who had DHF (n = 20, 60.6%) was higher in those

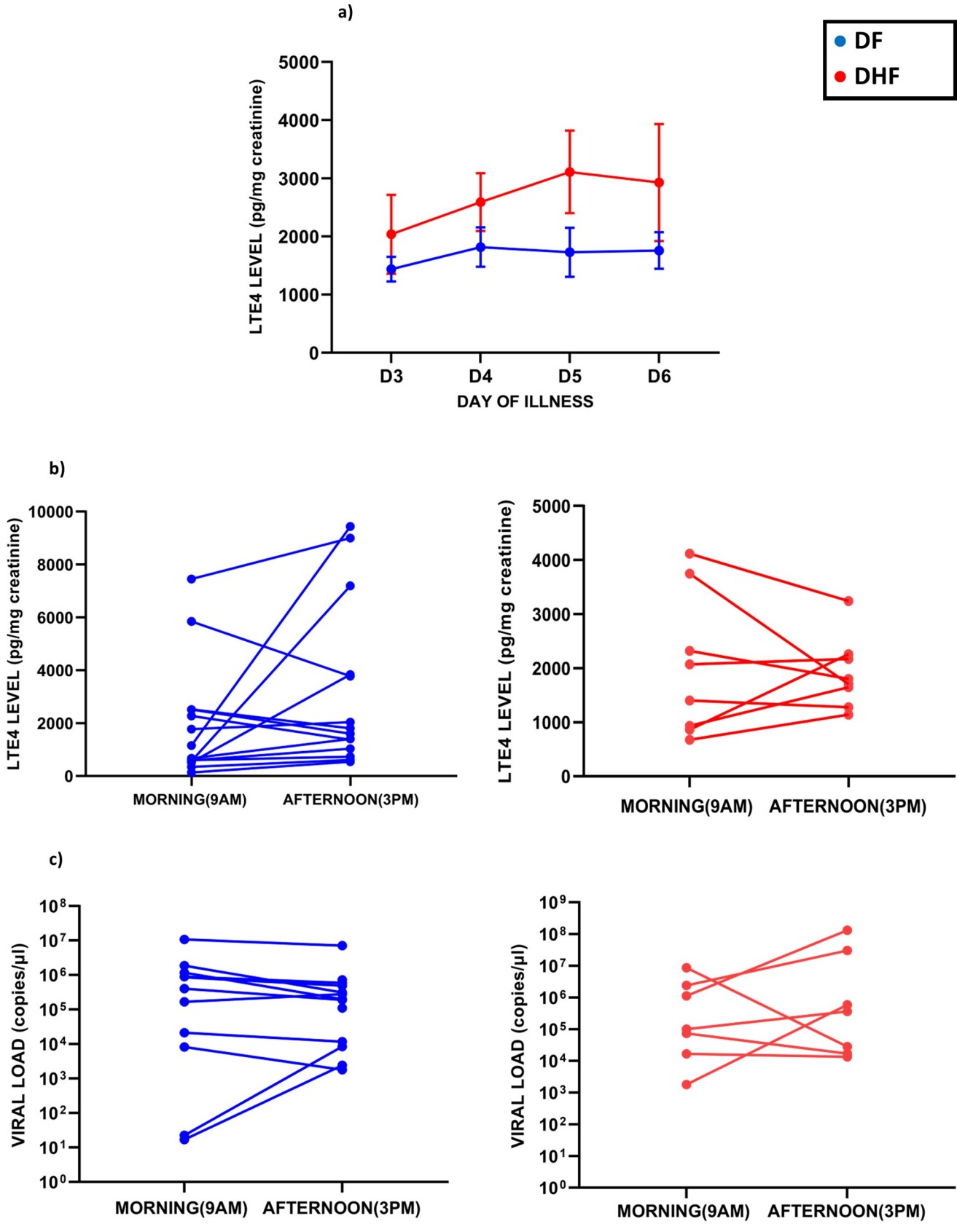

**Fig 4. Kinetics and diurnal changes of urinary LTE4.** Urinary LTE4 levels were measured daily from the day of admission to discharge in patients with DF (n = 25) and DHF (n = 23), throughout the course of illness. The blue lines represent DF and red DHF. Error bars represent standard error of mean (SEM) and mean (a). Urinary LTE4 levels (b) were measured in patients with DF (n = 14) and DHF (n = 8) and serum viral loads were measured in patients with DF (n = 13) and DHF (n = 7) (c). Changes in the LTE4 levels throughout the course of illness was compared using the Holm-Sidak method. The Wilcoxon paired signed rank test was carried out to determine significant of changes of LTE4, histamine and viral loads, in paired samples obtained in the morning and afternoon.

who were ≥ 41 years, compared to those who were ≤ 25 years (n = 16, 29.1%), which could have accounted for these changes. (**Fig 5C**).

## Urinary histamine levels in patients with acute dengue

A study carried out in a small cohort of patients, who had severe dengue showed that 24-hour urine histamine was higher in patients with DHF compared to healthy individuals [29].

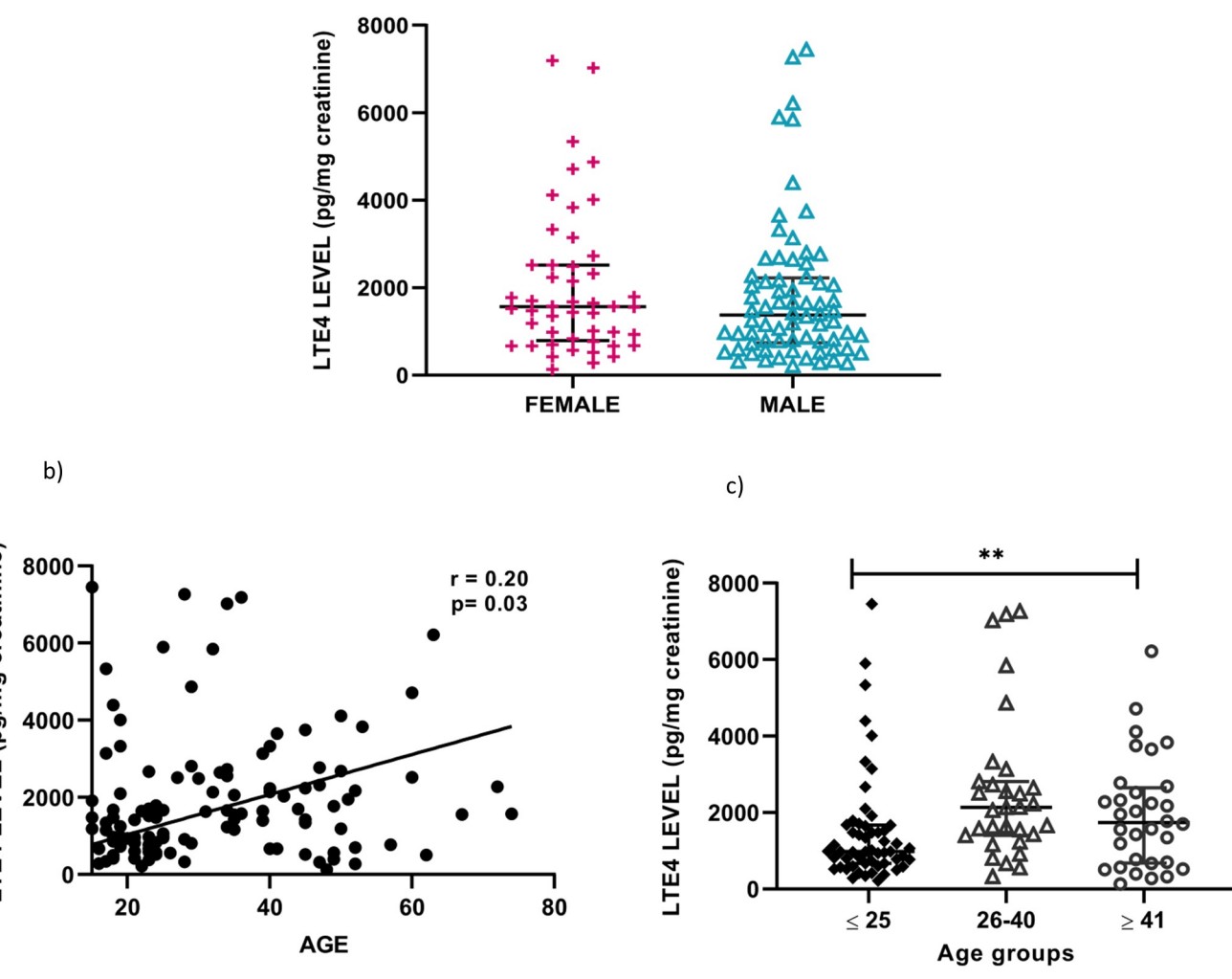

**Fig 5. Association of urinary LTE4 levels with age and gender in acute dengue.** Urinary LTE4 levels in patients with acute dengue were compared between females (n = 48) and males (n = 72) (a), correlated with the age of the individuals by Spearmans correlation coefficient (b), and compared between different age groups; those ≤ 25 years (n = 55), 26 to 40 years (n = 31) and ≥ 41 years (n = 32) (c). The comparisons were done using the Mann-Whitney test and Kruskal-Wallis test. Error bars represent the median and IQR. **P = <0.01.

Therefore, in order to understand the possible role of histamine in disease pathogenesis, we determined the urinary histamine levels in patients recruited ≤ 4 days of illness (n = 91), between 5 to 6 days of illness (n = 37) and 29 healthy individuals. Although, urinary histamine levels were assayed in all patients who were included in the urinary LTE4 analysis, we had to exclude samples of some patients due to inaccurate results even after repeated testing. Of the 91 patients recruited ≤ 4 days of illness, 52 had DF and 39 had DHF. In comparison to healthy individuals (median 0.22, IQR 0.07–0.76ng/mg creatinine), patients with DF (median 2.92, IQR 0.53–7.83ng/mg creatinine,) and those who proceeded to develop DHF (median 5.4, IQR 1.10–5.4ng/mg creatinine, p = < 0.0001) had significantly higher levels of urinary histamine (p<0.0001) (**Fig 6A**). Although patients with DHF had higher levels of urinary histamine compared to patients with DF it was not statistically significant (p = 0.31) (**Fig 6A**). Although the urinary histamine levels were lower during day 5 to 6 in patients with DF (median 0.15, IQR 0.06–0.86ng/mg creatinine, p = 0.80) compared to healthy individuals, the histamine levels of those with DHF was still high (median 0.29, IQR 0.05–1.12ng/mg creatinine). (**Fig 6A**).

There was no significant difference in urinary histamine levels in patients with primary and secondary dengue (**Fig 6B**) and there was no relationship with the viral loads (**Fig 6C**). The urinary histamine levels did not significantly correlate with urinary LTE4 levels in patients with DF (Spearman r = -0.08335, p = 0.56) or DHF (Spearman r = -0.2446, p = 0.13) (**Fig 6D**).

## Kinetics of changes of histamine throughout the course of illness

In order to determine the changes in urinary histamine throughout the illness, we measured histamine in 16 patients DF and 16 patients with DHF throughout the course of illness (**Fig 7A**). In patients with both DF and in those who proceeded to develop DHF, urinary histamine levels gradually declined from illness day 3 onwards. We also investigated possible diurnal variation in urinary histamine levels in 9 patients with DF and 7 patients with DHF (**Fig 7B**). In patients with both DF (p = 0.004) and DHF (p = 0.01) the urinary histamine levels significantly decreased in the afternoon (3pm) samples compared to the morning samples, although there was wide inter-individual variation (**Fig 7B**). Similar diurnal variations were seen with serum viral loads in patients with both DF and DHF (**Fig 7C**).

## Association of urinary histamine levels with age and gender in acute dengue

Of the 91 patients who were recruited during early illness (≤ 4 days of illness), 36 (40%) were females and 55 (60%) were males. There was no significant difference in the LTE4 levels between males and females (p = 0.78) or any correlation between age and histamine levels in acute dengue.

## Discussion

In this study we show that urinary LTE4 levels were significantly higher in early illness in patients, who proceed to develop DHF. The LTE4 levels continued to be higher during later stages of illness (day 5 to 6), in patients with DF and DHF and the levels during later stages of illness in patients with DHF were higher than those of early illness. Recently it was shown that serum chymase was a predictive biomarker of DHF and it increased throughout the course of illness in children who developed DHF [30]. Chymase is a mast cell product and since higher chymase levels were seen in the critical phase suggests increased mast cell degranulation occurs in the critical phase compared to early illness. Another mast cell product tryptase has also been shown to associate with dengue disease severity and increase vascular permeability of endothelial cells, and polymorphisms in the alpha-tryptase was associated with disease severity,

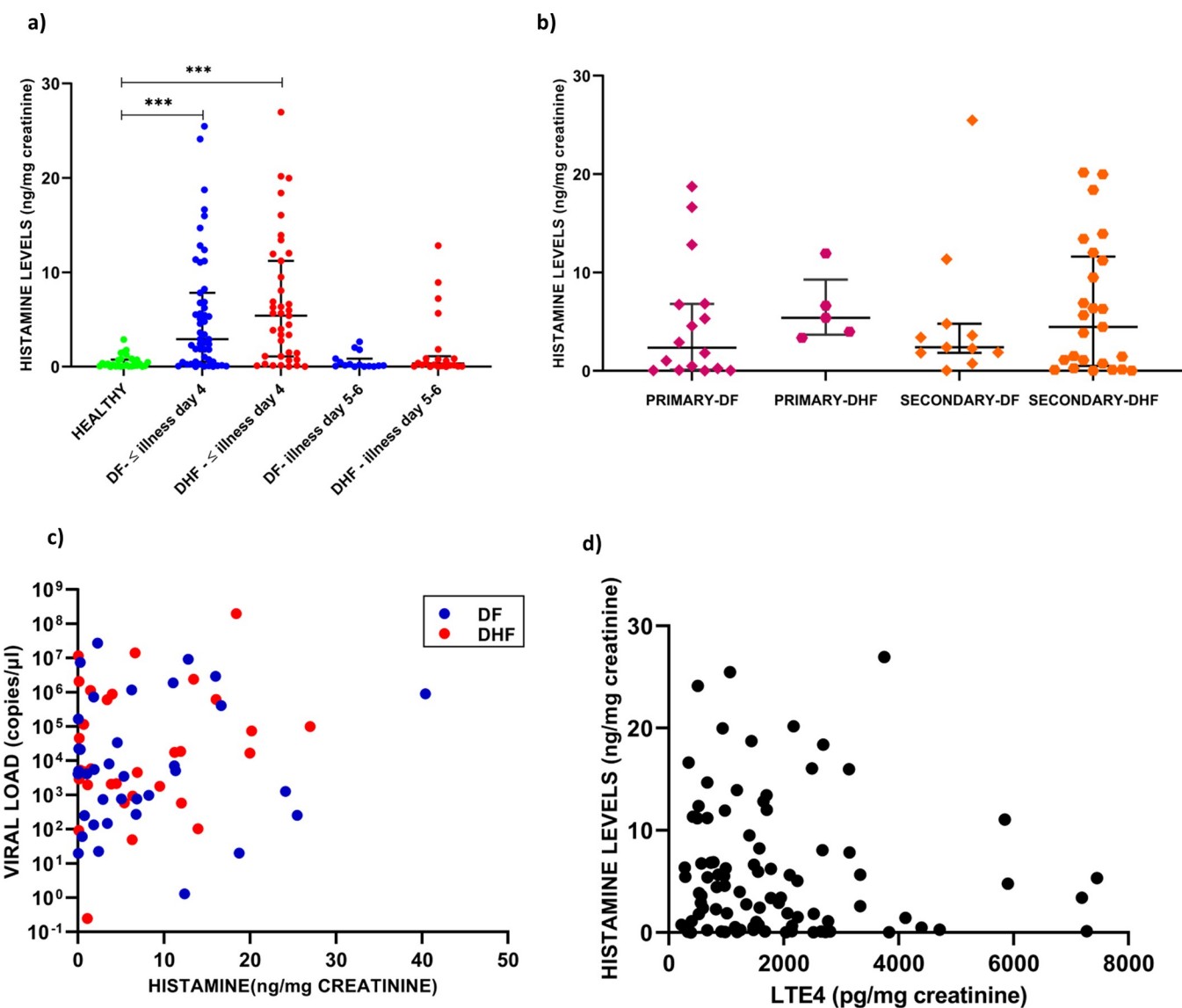

**Fig 6. Urinary histamine levels in acute dengue infection.** Urinary histamine levels were measured by quantitative ELISA, in healthy individuals (n = 29), in those with DF (n = 52) and those who progressed to develop DHF (n = 39) with <4 days of illness and in patients with DF (n = 15) and DHF (n = 25) in 5 to 7th day of illness (a). The histamine levels were compared patients with DF (n = 17) or DHF (n = 05) due to primary infection or DF (n = 11) or DHF (n = 25) due to secondary infection (b), were correlated with urinary LTE4 levels (c), correlated with the viral loads (d)The p values were calculated by the Mann-Whitney test and the correlations carried out by the Spearmans correlation. Error bars represent the median and IQR. *** P<0.0001.

suggesting that tryptase could be playing an important role in disease pathogenesis [23, 31]. Mast cells are the main producers of histamine and cysteinyl leukotrienes and therefore, as many studies have shown that other mast cell products are associated with clinical disease severity, it would be useful to determine if a stable urinary biomarker such as LTE4, could be used to predict development as DHF, as it is a convenient sample. However, although the levels of urinary LTE4 were significantly higher in those who proceeded to develop DHF during early illness, urinary LTE4 levels did not appear to perform well as a biomarker due to the low AUC values. This is possibly due to the wide diurnal variation seen in the LTE4 levels in patients with both DF and DHF.

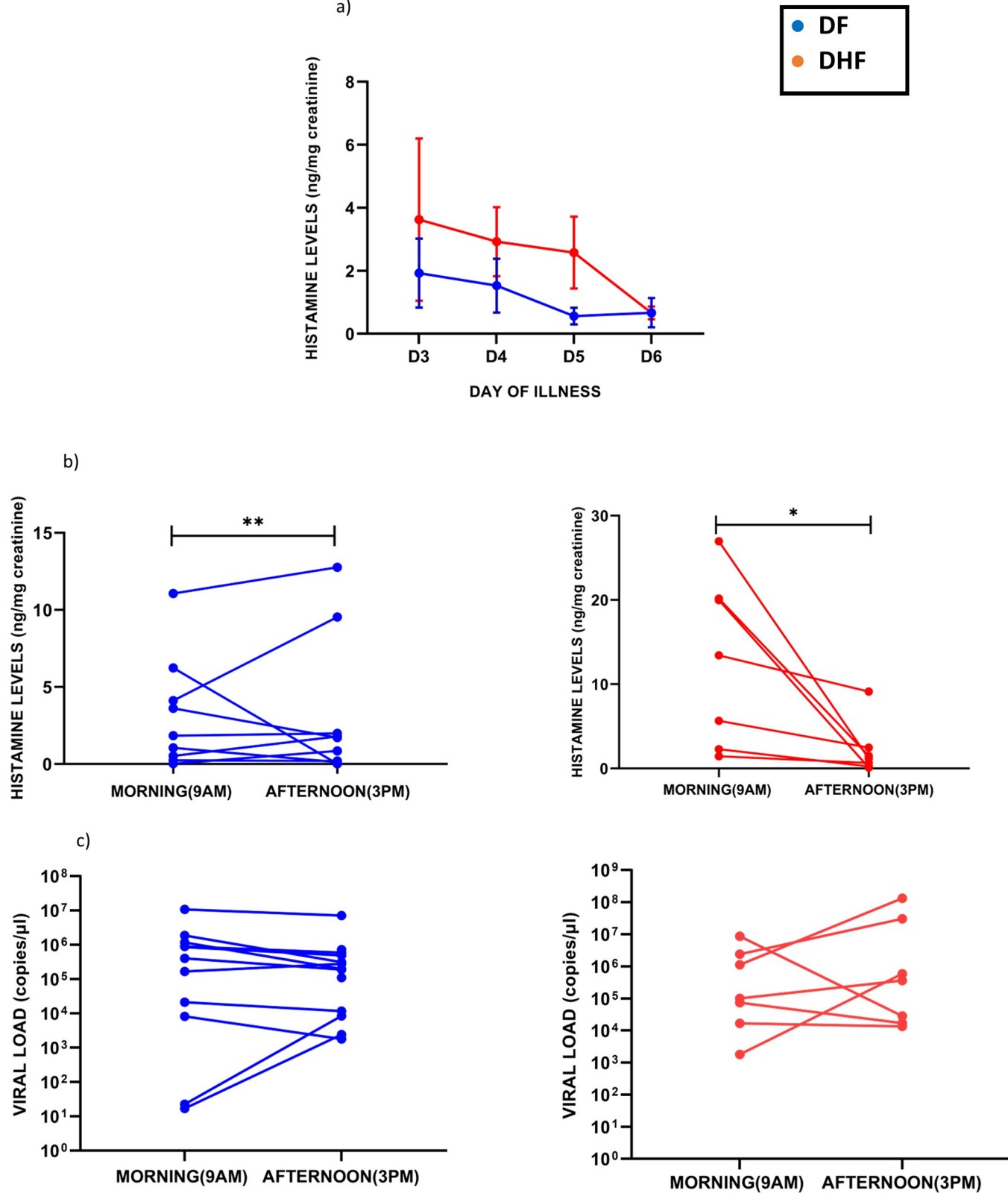

**Fig 7. Kinetics and diurnal changes of urinary histamine.** Urinary histamine levels were measured daily from the day of admission to discharge in patients with DF (n = 16) and DHF (n = 16), throughout the course of illness. The blue lines represent DF and red DHF. Error bars represent standard error of mean (SEM) and mean (a). Urinary histamine levels (b) were measured in patients with DF (n = 19) and DHF (n = 7) and serum viral loads were measured in patients with DF (n = 8) and DHF (n = 7) (c). *P <0.05.

Dirunal variation in both urinary LTE4 and histamine levels have been previously observed in patients with asthma and in normal individuals, although no specific pattern was observed, as in this study [32]. We too did not observe a consistent pattern in the changes of urinary LTE4 in the morning compared to the afternoon. This is in contrast with changes observed with PAF, with the levels always being high in the early morning levels, compared to the afternoon levels [9, 13]. Interestingly, similar changes were observed with viral loads between morning and afternoon samples, with a a 10-fold to 100-fold difference in some patients. Again, there was no particular pattern with the viral loads rising in some patients, while decreasing others and the changes in the viral loads did not show any relationship with the LTE4 or histamine levels, although the lack of any relationship could be due to the relatively smaller number of patients included in the analysis of the diurnal variation and kinetics of these markers. Many processes in the innate and adaptive immune system have shown to vary according to circardian rhythms including expression and function of toll like receptors [33], trafficking of monocytes [34] and secretion of cytokines and chemokines [35]. Therefore, the effect of the circadian rhythms which influence the immune response through glucocorticoids and neural pathways, should be further investigated [35].

Cysteinyl leukotrienes are synthesized from cells which posses LTC4 synthase, such as eosinophils, mast cells, basophils and macrophages [36]. Many different enzymes act on the subsequent products LTC4, LTD4 and subsequently LTE4, of which the stable N-acetyl derivate is found in the urine [36]. All these cells are readily infected by the DENV and are susceptible to enhanced infection and activation by the DENV in the presence of IgG antibodies which bind to the activating type of FcγRIIIA receptors [28, 37]. Indeed it was shown that mast cells were significantly more activated in secondary dengue infections [12]. Although LTE4 levels were not high in patients with secondary dengue infection, in this study, the levels were higher in those with DHF due to secondary dengue compared to those with DF. However, our results showed that overall the LTE4 levels were higher very early in illness, irrespective of serostatus and the degree of serum viraemia. Therefore, LTE4 production appears to be enhanced very early during infection, and increasing in those who develop DHF during the critical phase, suggesting that the pathway may contribute to the vascular leak. Montelukast is a cysteinly leukotriene receptor antagonist and is widely used to inhibit the action of leukotrienes in asthma and many other allergic diseases [38, 39]. In dengue mouse models it was shown to reduce the extent of vascular leak [12]. As montelukast is a relatively safe drug, it would be important to evaluate its efficacy in reducing dengue disease severity.

Mast cells are the main source of histamine although it can also be produced by basophils and the intestinal epithelium and other cells [40]. Histamine has been shown to induce mast cells to release leukotrienes, cytokines and also increase vascular permeability [40]. Although cross linking of the IgE bound to the high affinity FcεRI is the main trigger of histamine release from mast cells, complement components C3a, C5a and cytokines such as IL-33, IL-3, IL-18 and GM-CSF have been shown to induce synthesis and release [41]. Although we found that histamine levels were significantly higher in patients with acute dengue, there was no difference in their levels in those with DF compared to those with DHF. In addition, the histamine levels fell in both patients with DF and DHF during the course of illness, which was in contrast to what was observed with urinary LTE4 levels. Interestingly, unlike LTE4 levels, the urinary histamine levels were always lower in the afternoon when compared to the morning, as seen with the variation of PAF [9, 13]. We recently conducted a phase II, randomized, placebo controlled clinical trial with rupatadine to evaluate its efficacy in reducing vascular leak in acute dengue [3]. Rupatadine significantly inhibited the effects of dengue sera on human endothelial cells lines by reducing the reduction in trans-endothelial resistance and a reduction is ZO-1 expression and also showed small but significant differences in the extent of fluid leakage and

thrombocytopenia, when the drug was given early [3]. Rupatadine is an antihistamine with PAF receptor antagonist effects [42]. Therefore, it would be important to evaluate if some of the effects of rupatadine was also due to its antihistamine effects.

## Conclusions

We found that urinary LTE4 levels were significantly higher in early illness in patients, who proceed to develop DHF. The LTE4 levels continued to rise during the course of illness in patients with DHF, whereas the levels remained unchanged in those with DF. As there are cysteinyl leukotriene receptor antagonists available, it would be imporant to evaluate if these drugs could reduce dengue disease severity.

## Author Contributions

**Conceptualization:** Graham S. Ogg, Gathsaurie Neelika Malavige.

**Data curation:** Tehani Silva, Chathurika Gangani, Sameera D. Mahapatuna, Ananda Wijewickrama.

**Formal analysis:** Tehani Silva, Gathsaurie Neelika Malavige.

**Funding acquisition:** Graham S. Ogg, Gathsaurie Neelika Malavige.

**Investigation:** Tehani Silva, Laksiri Gomes, Chathurika Gangani, Sameera D. Mahapatuna, Thilagaraj Pathmanathan, Ananda Wijewickrama.

**Methodology:** Laksiri Gomes.

**Project administration:** Chandima Jeewandara.

**Resources:** Chandima Jeewandara.

**Supervision:** Chandima Jeewandara, Gathsaurie Neelika Malavige.

**Validation:** Laksiri Gomes.

**Writing – original draft:** Tehani Silva, Gathsaurie Neelika Malavige.

**Writing – review & editing:** Graham S. Ogg, Gathsaurie Neelika Malavige.

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
