## [Decision Letter · Decision Letter 0]

19 Nov 2020

PONE-D-20-29451

Urinary leukotrienes and histamine in patients with varying severity of acute dengue

PLOS ONE

Dear Dr. Neelika,

Thank you for submitting your manuscript to PLOS ONE. After careful consideration, we feel that it has merit but does not fully meet PLOS ONE’s publication criteria as it currently stands. Therefore, we invite you to submit a revised version of the manuscript that addresses the points raised during the review process.

Please answer all the comments and include the sharing information as requested

We look forward to receiving your revised manuscript.

Kind regards,

Eliseo A Eugenin, Ph.D.

Academic Editor

PLOS ONE

Journal Requirements:

https://journals.plos.org/plosntds/article?id=10.1371%2Fjournal.pntd.0003459

In your revision ensure you cite all your sources (including your own works), and quote or rephrase any duplicated text outside the methods section. Further consideration is dependent on these concerns being addressed.

Additional Editor Comments (if provided):

Dear Dr. Neelika

Thank you for submit your manuscript to PLOSone. Please answer all the comments and also provide the information than all the data will be available

Best Regards

Eliseo

Reviewers' comments:

Reviewer's Responses to Questions

**Comments to the Author**

1. Is the manuscript technically sound, and do the data support the conclusions?

Reviewer #1: Yes

Reviewer #2: Yes

2. Has the statistical analysis been performed appropriately and rigorously? 

Reviewer #1: Yes

Reviewer #2: Yes

3. Have the authors made all data underlying the findings in their manuscript fully available?

Reviewer #1: No

Reviewer #2: Yes

4. Is the manuscript presented in an intelligible fashion and written in standard English?

Reviewer #1: Yes

Reviewer #2: Yes

5. Review Comments to the Author

Reviewer #1: This is an interesting and well written report describing the kinetics of urinary LTE4 and histamine in DF versus DHF patients. This information is particularly interesting because it appears to confirm some of the animal model research that described a role for these factors in dengue disease. The authors also appropriately emphasize how their data shows limitations towards the application of these biomarkers for disease prognosis, for example by including a ROC analysis, which is a strong aspect of the study. Some improvements and corrections are needed before publication.

Specific comments:

When values for a subset of patients are reported, for example for the longitudinal sampling, it would be important to explain in the methods how subsets of patients were selected.

Even if possibly the same citation, 12, the last sentence on p. 4 contains very specific info and the citation should be located there as well to be clear.

“US” in table 1 should be spelled out for clarity and considering the multidisciplinary journal audience.

“likely to be due to increased infection of mast cell” may not be correct because mast cells have not been shown to be infected in humans. I think it should just read increased degranulation of mast cells, since the biomarkers are a degranulation product.

p. 4- For the section heading “Usefulness of LTE4” and the corresponding Fig. 3 legend, it should be updated to urinary LTE4 for clarity since serum levels weren’t assessed and this could lead to confusion since serum is often used in biomarker studies.

“urinary LTE4 levels rose in patients with DHF… (Fig 4a)” – It’s not clear if they were patients who had already been diagnosed with DHF or if some had not yet?

I think the citation Syenina et al 2015 for the sentence on p. 17 under the heading of urinary histamine may not be the correct one.

Can the authors clarify if the LTE4 and histamine were measured in the same patient cohort? Any discrepancies in the numbers if it is the same cohort should be explained.

The persistence of urinary histamine and LTE4 in DHF patients is quite interesting since it was also very recently shown in a longitudinal study of pediatric patients with severe dengue in Sri Lanka that chymase is not only higher at acute time points but also persists longer in DHF patients. (doi: 10.1038/s41598-020-68844-z.) This might be worth discussing.

Perhaps the Figure 2 legend title should be changed since there was no association found (right? No signs of significance on graphs), but the title implies otherwise. Also please check the panel order. Statistical tests for viral load correlation need to be written in the figure legend and the trend lines on the graph along with the Pearson’s R value. Similarly for Fig. 3, can the authors put the AUC on the graph?

Figure 4 doesn’t have any indication of how statistical analysis was performed in the figure legend.

Reviewer #2: This is an interesting manuscript that builds on previous observations on inflammatory mediator production in Dengue.

1. The authors should more clearly indicate that they are measuring an LTE4 metabolite and not LTE4 itself in their urinary analysis

2. Observations regarding diurnal variation might be better considered in the context of endogenous cortisol levels. This should at least be commented on.

3. It is surprising that there is little indication that there is little difference between severe and more mild dengue disease for some of the markers analysed. Given the variation in readouts, does the study have sufficient power to reach this conclusion? The authors should justify their sample size.

4. Several studies have suggested that increased mast cell activation and increased plasma tryptase are associated with severe dengue disease. The authors should consider these data relative to these studies in more detail. For example, are they suggesting that observed histamine and cystienyl leukotriene responses are not primarily from mast cells where these data are not consistent with tryptase measurements. Are the urinary measurements they provide giving a more consistent picture of such mediator production and cellular activation than one time measures of plasma markers in other studies?

5. To what extent could disease associated changes in kidney function alter the levels of mediators being examined in the urine? Further commentary on this issue would be helpful, beyond the data provided.

6. Further evaluation of the levels of histamine and LTE4 metabolite in urine relative to time of onset of disease within each of the different disease severity groups might be useful for interpretation

6. PLOS authors have the option to publish the peer review history of their article (what does this mean?). If published, this will include your full peer review and any attached files.

Reviewer #1: No

Reviewer #2: No

---

## [Author Response · Author response to Decision Letter 0]

15 Dec 2020

Reviewer 1

Thank you for giving me the opportunity to review the manuscript.

This is an interesting and well written report describing the kinetics of urinary LTE4 and histamine in DF versus DHF patients. This information is particularly interesting because it appears to confirm some of the animal model research that described a role for these factors in dengue disease. The authors also appropriately emphasize how their data shows limitations towards the application of these biomarkers for disease prognosis, for example by including a ROC analysis, which is a strong aspect of the study. Some improvements and corrections are needed before publication.

1) When values for a subset of patients are reported, for example for the longitudinal sampling, it would be important to explain in the methods how subsets of patients were selected.

Response: We apologize for the lack of clarity regarding the subsets of patients included in the study. We have given details regarding the recruitment of patients to determine kinetics throughout the illness and diurnal variations in LTE4 and viral loads in patients, in the revised version of the manuscript.

2) Even if possibly the same citation, 12, the last sentence on p. 4 contains very specific info and the citation should be located there as well to be clear.

Response: We have included citation in the revised manuscript. 

3) “US” in table 1 should be spelled out for clarity and considering the multidisciplinary journal audience. 

Response: ‘US” has been replaced with ‘ultrasound’ in the revised manuscript. 

4) “likely to be due to increased infection of mast cell” may not be correct because mast cells have not been shown to be infected in humans. I think it should just read increased degranulation of mast cells, since the biomarkers are a degranulation product.

Response: We thank the reviewer for this suggestion, and we have corrected this in the revised version of the manuscript. 

5) p. 4- For the section heading “Usefulness of LTE4” and the corresponding Fig. 3 legend, it should be updated to urinary LTE4 for clarity since serum levels weren’t assessed and this could lead to confusion since serum is often used in biomarker studies.

Response: We apologize for not including “urinary” in the fig 3 legend. We have included it in the revised manuscript. 

6) urinary LTE4 levels rose in patients with DHF… (Fig 4a)” – It’s not clear if they were patients who had already been diagnosed with DHF or if some had not yet?

Response: We apologize for the lack of clarity. At the time of recruitment of those with a duration of illness of ≤ 4 days of illness, none had evidence of fluid leakage. A subset of these 120 patients recruited during early illness, were included in determining the kinetics. More details regarding recruitment and the absence of fluid leakage at the time of recruitment is included in the revised version of the manuscript. 

7) I think the citation Syenina et al 2015 for the sentence on p. 17 under the heading of urinary histamine may not be the correct one. 

We are grateful to the reviewer and have amended this in the revised manuscript and included the correct citation Tuchinda et al 1977.

8) Can the authors clarify if the LTE4 and histamine were measured in the same patient cohort? Any discrepancies in the numbers if it is the same cohort should be explained.

Response: We thank the reviewer for this question. All patients in whom the histamine was analysed were those who were recruited to investigate urinary LTE4 study levels. However, we had to exclude histamine levels from some patients as we could not get accurate results from the ELISA (too high or minus values), even after repeating these samples. We have included an explanation in the revised version of the manuscript. 

9) The persistence of urinary histamine and LTE4 in DHF patients is quite interesting since it was also very recently shown in a longitudinal study of pediatric patients with severe dengue in Sri Lanka that chymase is not only higher at acute time points but also persists longer in DHF patients. (doi: 10.1038/s41598-020-68844-z.) This might be worth discussing.

Response: We thank the reviewer for this suggestion. We have discussed the recent findings from the above-mentioned study in the discussion section of the revised manuscript. 

10) Perhaps the Figure 2 legend title should be changed since there was no association found (right? No signs of significance on graphs), but the title implies otherwise. Also please check the panel order. Statistical tests for viral load correlation need to be written in the figure legend and the trend lines on the graph along with the Pearson’s R value. Similarly, for Fig. 3, can the authors put the AUC on the graph?

Response: We thank the reviewer for all these suggestions. We have described the data displayed in the 2 and 3 in the figure legend and have deleted the p values from the figure 2 legend. We have made changes to figure legend titles, figure legends and add statistical values as per the suggestions in the revise version of the manuscript.

11) Figure 4 doesn’t have any indication of how statistical analysis was performed in the figure legend.

Response: We apologies for this omission. We have included the statistical information in the revised version of the manuscript. 

Reviewer 2

This is an interesting manuscript that builds on previous observations on inflammatory mediator production in Dengue.

1) The authors should more clearly indicate that they are measuring an LTE4 metabolite and not LTE4 itself in their urinary analysis.

Response: We thank the reviewer for this question. We used Leukotriene E4 ELISA kit (Cayman chemical) for the measurement of urinary LTE4 levels, it is the LTE4 itself that was measured as it is the stable end product of cysteinyl leukotriene and is not further modified. To clarify this matter, we made relevant amendments to the introduction of the revised manuscript. 

2) Observations regarding diurnal variation might be better considered in the context of endogenous cortisol levels. This should at least be commented on.

Response: We thank the reviewer for this important suggestion, which we have included in the discussion part of the revised manuscript. 

3) It is surprising that there is little indication that there is little difference between severe and more mild dengue disease for some of the markers analyzed. Given the variation in readouts, does the study have sufficient power to reach this conclusion? The authors should justify their sample size.

Response: We thank the reviewer for this comment. Although we have investigated urinary LTE4 and histamine levels during early and late illness in a large cohort, the changes of these markers throughout the course of illness and the diurnal variations were only assessed in a smaller cohort due to difficulties in obtaining samples throughout the course of illness and twice a day in patients who are bled for their routine clinical management several times a day. We have included this limitation in the discussion. 

4) Several studies have suggested that increased mast cell activation and increased plasma tryptase are associated with severe dengue disease. The authors should consider these data relative to these studies in more detail. For example, are they suggesting that observed histamine and cystienyl leukotriene responses are not primarily from mast cells where these data are not consistent with tryptase measurements? Are the urinary measurements they provide giving a more consistent picture of such mediator production and cellular activation than one-time measures of plasma markers in other studies?

Response: We thank the reviewer for this important comment. Based on the study data, we do not have any evidence to show that sole source of histamine or LTE4 are mast cells, as especially LTE4 is known to be produced by other cells as well. However, we have mentioned that they are likely to be from mast cells given that histamine is produced predominantly by mast cells. 

5) To what extent could disease associated changes in kidney function alter the levels of mediators being examined in the urine? Further commentary on this issue would be helpful, beyond the data provided.

Response: We thank the reviewer for this question. The kidney function can alter the levels of the mediators being investigated and for both LTE4 and histamine. We have accordingly normalized it to the urinary creatinine levels as described in methods. We have further elaborated on this in the revised manuscript. 

6) Further evaluation of the levels of histamine and LTE4 metabolite in urine relative to time of onset of disease within each of the different disease severity groups might be useful for interpretation. 

Response: We thank the reviewer for this important suggestion. We have accordingly evaluated the urinary LTE4 and histamine levels during early illness and late illness and investigated their changes throughout the course of illness and their diurnal variation in a subset of patients. Analyzing the data for different severity groups, for each day of the illness is not sufficiently powered in this cohort for accurate assessment.

---

## [Decision Letter · Decision Letter 1]

11 Jan 2021

Urinary leukotrienes and histamine in patients with varying severity of acute dengue

PONE-D-20-29451R1

Dear Dr. Gathsaurie,

We’re pleased to inform you that your manuscript has been judged scientifically suitable for publication and will be formally accepted for publication once it meets all outstanding technical requirements.

Kind regards,

Eliseo A Eugenin, Ph.D.

Academic Editor

PLOS ONE

Additional Editor Comments (optional):

Thank you for correcting the issues raised by the reviewers

Eliseo

Reviewers' comments:

Reviewer's Responses to Questions

**Comments to the Author**

1. If the authors have adequately addressed your comments raised in a previous round of review and you feel that this manuscript is now acceptable for publication, you may indicate that here to bypass the “Comments to the Author” section, enter your conflict of interest statement in the “Confidential to Editor” section, and submit your "Accept" recommendation.

Reviewer #1: All comments have been addressed

2. Is the manuscript technically sound, and do the data support the conclusions?

Reviewer #1: Yes

3. Has the statistical analysis been performed appropriately and rigorously? 

Reviewer #1: Yes

4. Have the authors made all data underlying the findings in their manuscript fully available?

Reviewer #1: Yes

5. Is the manuscript presented in an intelligible fashion and written in standard English?

Reviewer #1: Yes

6. Review Comments to the Author

Reviewer #1: The authors could have more insightfully discussed the other work on dengue biomarkers that was requested by both reviewers, but nevertheless I consider the major comments to have been addressed.

7. PLOS authors have the option to publish the peer review history of their article (what does this mean?). If published, this will include your full peer review and any attached files.

Reviewer #1: No

---

## [Editor Report · Acceptance letter]

27 Jan 2021

PONE-D-20-29451R1 

Urinary leukotrienes and histamine in patients with varying severity of acute dengue 

Dear Dr. Malavige:

I'm pleased to inform you that your manuscript has been deemed suitable for publication in PLOS ONE. Congratulations! Your manuscript is now with our production department. 

Kind regards, 

on behalf of

Dr. Eliseo A Eugenin 

Academic Editor

PLOS ONE